# Reversal of Osseointegration as a Novel Perspective for the Removal of Failed Dental Implants: A Review of Five Patented Methods

**DOI:** 10.3390/ma14247829

**Published:** 2021-12-17

**Authors:** Rolf G. Winnen, Kristian Kniha, Ali Modabber, Faruk Al-Sibai, Andreas Braun, Reinhold Kneer, Frank Hölzle

**Affiliations:** 1Private Practice, 40217 Düsseldorf, Germany; 2Department of Oral and Maxillofacial Surgery, RWTH Aachen University Hospital, 52074 Aachen, Germany; kkniha@ukaachen.de (K.K.); amodabber@ukaachen.de (A.M.); fhoelzle@ukaachen.de (F.H.); 3Institute of Heat and Mass Transfer, RWTH Aachen University, 52056 Aachen, Germany; kneer@wsa.rwth-aachen.de; 4Clinic for Operative Dentistry, Periodontology and Preventive Dentistry, RWTH Aachen University Hospital, 52074 Aachen, Germany; anbraun@ukaachen.de

**Keywords:** reversibility of osseointegration, dental implants, explantation, implant removal, failed implants, reimplantation

## Abstract

Osseointegration is the basis of successful dental implantology and the foundation of cementless arthroplasty and the osseointegrated percutaneous prosthetic system. Osseointegration has been considered irreversible thus far. However, controlled heating or cooling of dental implants could selectively damage the bone at the bone–implant interface, causing the reversal of osseointegration or “osseodisintegration”. This review compares five methods for implant removal, published as patent documents between 2010 and 2018, which have not yet been discussed in the scientific literature. We describe these methods and evaluate their potential for reversing osseointegration. The five methods have several technical and methodological similarities: all methods include a handpiece, a connecting device for coronal access, and a controlling device, as well as the application of mechanical and/or thermal energy. The proposed method of quantifying the temperature with a sensor as the sole means for regulating the process seems inadequate. A database used in one of the methods, however, allows a more precise correlation between a selected implant and the energy needed for its removal, thus avoiding unnecessary trauma to the patient. A flapless, microinvasive, and bone-conserving approach for removing failed dental implants, facilitating successful reimplantation, would benefit dental implantology. These methods could be adapted to cementless medical implants and osseointegrated percutaneous prosthetics. However, for some of the methods discussed herein, further research may be necessary.

## 1. Introduction

The increasing use of dental implants and increasing implant function times have made implant failures unavoidable. Peri-implantitis (81.9%), implant malpositioning (2–14%, correlated with the experience of the surgeon), and implant fracture (1%) are frequent causes of implant failure [1] and indications for implant removal. Peri-implantitis and malpositioning are also relevant from an aesthetic perspective, which is integral to dental implant treatment [2]. Peri-implantitis is a pathological condition characterized by peri-implant inflammation and bone loss. It progresses faster than periodontitis and shows an accelerating pattern [3]. The term peri-implantitis yielded almost 3000 search results on PubMed, with over 400 in 2020 alone, and it is undoubtedly a matter of contention. Its prevalence depends on various factors, but the current literature indicates a two-digit percentage range at the implant level after 10 years of function [4,5,6]. The criteria for the preservation or removal of implants are not yet clearly defined [2] and remain a subject requiring further research [7]. Peri-implantitis can be treated to an extent to preserve the implant [8,9,10]. However, there is consensus that implants with severe, progressing peri-implantitis should be removed [1,2,7,11,12,13,14,15,16,17]. It has been established that periodontitis negatively impacts systemic health [18,19,20,21]. Since there are similarities between periodontitis and peri-implantitis, it is expected that recent studies also suggested peri-implantitis as having negative impacts on systemic health [22,23,24,25]. The general concept regarding the interaction between systemic conditions and peri-implantitis has been described in several papers [26,27,28]. These data emphasize the necessity of removing implants with severe, progressing peri-implantitis.

Methods such as the counter-torque ratchet technique (CTRT), dental extraction kits, and conventional resective implant removal using burs or trephines have been described for the removal of implants [1,7,15]. As described by Froum et al. [14] and Anitua et al. [11,14,29,30,31], CTRT involves the application of a counterclockwise torque to the implant. CTRT is considered the most conservative technique. However, it has clear limitations. This technique requires a high force, which may fracture the peri-implant bone [32]. Braegger et al. stated that “unscrewing is successful only when the implant is apically integrated only a few millimeters” [32] (p. 1). It has been reported that a maximum remaining osseointegration of 4 mm is critical when using CTRT alone [1,33]. This value is further influenced by the quality of the residual osseointegrating bone.

Osseodisintegration can be considered the antonym of osseointegration. The term osseodisintegration was coined by Tonetti in 1998 [34] to describe the dissolution of osseointegration caused by peri-implant inflammation or overloading, which is unintentional and undesirable. Solderer first used this term in 2019 to describe the intentional dissolution of osseointegration in the context of implant removal [1]. The concept of osseointegration has been applied in medicine for decades. However, the intentional reversal of the osseointegration process has not yet been demonstrated in routine clinical practice. Therefore, the concept of reversibility of osseointegration has not yet been established.

There are extensive descriptions of the deleterious effects of heat on bone in the literature. In 1982 and 1984 [35,36], two reports observed consistent and widespread bone tissue injury caused by exposure to a temperature of 50 °C for 1 min in a rabbit model using vital microscopy. In 1999, Li et al. observed that osteoblasts exposed to a temperature of 42 °C or 45 °C for 10 min recovered, while cells exposed to 48 °C did not [37]. The extent of bone damage depends on the temperature and duration of its application. Application of heat between 47 °C and 55 °C for 1 min can cause irreversible bone damage [38]. Cold cryoinsults also caused osteonecrosis in an emu bone model [38,39], with the moderated critical temperature of 3.5 °C for the reduction in osteocyte viability below 50%. The holding time and the rate of temperature change were not considered in the study [39].

A recent systematic review on preclinical in vivo research revealed no clear threshold values of hot and cold stimuli causing bone necrosis in the existing literature. The authors recommended in-depth clinical studies to gain further insight into the potential of thermal necrosis for implant removal [38]. An in vitro pilot study on the induction of thermal necrosis for implant removal showed significant degeneration of the bone matrix at 51 °C for 10 s and 5 °C for 30 s [40].

Implants can be osseointegrated completely even in the presence of malpositioning or implant fracture as indications for implant removal. The osseointegrated part of the implant can comprise considerably more than the apical 4 mm when moderate or medium peri-implantitis is the indication. In such cases, the implant must initially be separated from the bone to an extent such that CTRT becomes possible. The classical approach involves resection using a trephine bur. This can result in large bone defects, and the loss of the cross-sectional volume can limit the possibility of reimplantation, rendering significant bone augmentation necessary. Damage to the adjacent teeth and titanium particles produced during milling remaining in the tissue following the treatment are the other risk factors associated with the use of trephine burs [32]. The trephine bur must fit closely to the implant to minimize the resection of bone. In cases of tapering implants, it is recommended to reduce the cervical portion of the implant with a highspeed cutter first [32]. Cutting and trephining can increase the temperature of the bone, especially in the deeper regions, leading to uncontrolled bone necrosis [32,41]. Therefore, some experts consider the use of trephine burs in implant removal obsolete [17].

Piezosurgery can alternatively be used to remove the bone surrounding the implant [1]. Solderer et al. mentioned two new methods of implant removal. The first method is also resective and uses lasers instead of a trephine bur to remove the bone surrounding a failing endosseous dental implant [42]. Another study compared Er,Cr:YSGG laser (erbium, chromium: yttrium–scandium–gallium–garnet) to a trephine bur [43]. Both working groups concluded that lasers were superior to trephine burs in terms of bone preservation, minimal thermal damage, and greater cutting efficiency. However, the use of lasers was more time-consuming.

The second method is non-resective. It intentionally induces thermo-necrosis to loosen the implant from the bone using an electrosurgical probe as the source of heat [44,45]. Another paper described the use of dental lasers for the same purpose [46]. Both were case reports, and the procedures were only performed experimentally. This approach of applying thermal energy to an implant for the intentional dissolution of osseointegration at the bone–implant interface forms the basis of this review. The indications are the same as those for conventional implant removal. However, this technique is less invasive and more bone-conserving than resective implant removal.

This review focuses on five methods disclosed in the patent literature for the removal of failed dental implants that can be found in the Patentscope database of the World Intellectual Property Organization. The patents were selected on the basis of results of the search conducted by the European Patent Organization (EPO) in the Patentscope database during the patent application process of Patent 5 (see below). The searches were conducted according to “search procedure and strategy” described in Part B of the Guidelines for Examination in the EPO [47]. The purpose of this search strategy is to find all similar patents (in this case, devices for the removal of enossal implants through the application of energy). Since EPO is renowned for its diligence in this respect, this goal was achieved. The authors of these patents noted the disadvantages of the conventional techniques for implant removal and critically commented on the bone loss associated with resective implant removal using burs [32,48,49,50]. They set out to develop flapless, microinvasive, and bone-conserving approaches for implant removal to reduce trauma and facilitate early reimplantation, which led to the development of these methods to induce thermo-necrosis for intentional osseodisintegration.

For successful clinical application, predictable and reproducible approaches must be used for heating and/or cooling implants with different properties. The potential of these methods described in the patent literature has not yet been scientifically discussed. Herein, we review this literature to assess how the different methods address the problem of facilitating implant removal and allow the reversal of osseointegration. Some of these methods are important milestones in the ongoing research on the atraumatic removal of enossal implants.

In this review, the term “to temper” is used to signify “to bring to an intended temperature”.

## 2. Five Novel Patented Approaches for Implant Removal

### 2.1. Principles

The five methods discussed in this review are as follows:Patent 1 (P1): Device for removing dental implant fixtures (machine translation from the original Korean publication).Patent 2 (P2): Device for detachment and explantation of bone implants.Patent 3 (P3): Device for destroying a connection between an implant and biological tissue.Patent 4 (P4): Device for loosening, insertion, and removal of dental implants.Patent 5 (P5): Device for the controlled removal of osseointegrated implants and improved dental implants.

Only P1 and P5 are actual patents. In contrast, P2 through P4 are mere patent applications that seem to have not undergone patent examination before any Patent Office. Therefore, P2 through P4 are unexamined and may not necessarily disclose inventions in a legal sense, i.e., may not disclose patentable subject matter. Despite this clear legal difference, P1 through P5 are herein referred to as “patents” only for the sake of ease of reference. The patents (meaning patent documents) were published by the respective patent offices between 2011 and 2018. Three patents designate researchers affiliated with a university, while the other two were registered by practicing dentists. All methods were published as patent publications only, and Patent 1 was published in Korean. Hence, they are not listed in MEDLINE or other medical databases; furthermore, to the best of our knowledge, as of October 2021, these methods have not been discussed in the scientific literature. As these patents suggest interesting new perspectives relevant to the ongoing scientific research, their review as outlined herein seems of interest to the researching community. All mentioned patent publications can be found in the Appendix A.

### 2.2. P1 (Lee, Korea)

The central idea of this approach is the application of heat to the implant. The apparatus consists of a heating unit (120), a temperature sensor (210), and a control unit (Figure 1). The heating unit provides heat between 40 °C and 1000 °C, and the temperature sensor measures the temperature of the implant (20) and the surrounding tissues (10) via direct contact or infrared radiation. The heat applied to the implant is regulated by the control unit via sensor measurement data. A temperature between 40 °C and 60 °C, more precisely of 46 °C to 47 °C, is considered necessary in P1. Heat should be applied for a duration sufficient to induce osteonecrosis of the peri-implant bone cells. The controlling device has an alarm function that alerts the operator when a preset temperature or time value is reached. The application of heat can either be stopped immediately or continued at the operator’s discretion.

The handpiece (60) contains a heating unit and a connecting device (110), shaped based on a secondary part fitting into the internal geometry of the implant. The contact between the connecting device and the implant can be a point, line, or surface contact. The thermal conductivity may be increased using a highly thermal compound material (40) [51]. However, Figure 1 of P1 shows that the connecting device only reaches about halfway into the implant. As a result, uneven heating may occur in the implant, where the bottom of the implant, remote from the connecting device, is heated in a delayed fashion.

### 2.3. P2 (Braegger et al., Switzerland)

Braegger et al. subsumed the conventional methods of implant removal as “hard explantation” methods. They suggested the antonym “soft explantation” to describe a novel technique of gently loosening an implant from the bone with minimal damage to the adjacent tissues [32] (p. 1). The central idea behind this method involves the removal of implants using a combination of low temperature and mechanical forces.

The device consists of a handpiece containing a grip head (6) (Figure 2), heating or cooling module (7), vibration module (8), hammering module (9), handle (10), a secondary clamping system composed of a clamping lever (11), and clamping jaws (12). The grip head can exert radial clamping forces via an adapter piece (connecting device) on the implant (1) (Figure 3) in the bone (4). This configuration enables the user to manually rotate the handle around the axis of the implant.

The idea of cooling the implant is based on the freeze fracturing technique described by Donath [52]. Cooling the implant to a very low temperature leads to the fracture of the bone–implant interface. The device contains a receptacle in the cooling module that can be filled with dry ice or any other material that can be cooled externally before use. The cold temperature can also be achieved directly using a pressurized CO_2_ container, which can be attached to the handpiece. Alternatively, the cooling agent can be fed through an appropriate channel. The transfer of temperature from the cooling/heating module to the implant occurs through a connecting device. According to empirical tests carried out by Braegger et al., the induction of the cooling effect requires approximately 2 to 3 min. However, because of the use of a grip head (6), the cooling effect will commence only at the upper end of the implant and is, therefore, not distributed uniformly across the length of the implant. In fact, the cooling may take up to 3 min. Furthermore, a constant measuring of the heat/cold applied to the implant seems unavoidable in view of the disclosure in P2.

In addition to the application of cold, vibrational energy transmitted through the implant head can be used as a second stimulus to detach the implant from the bone. The vibration energy may be generated using an eccentric electric motor (6000–40,000 rpm), creating vibrations with a frequency of approximately 100–700 Hz that can be transmitted to the grip head tool. Higher-frequency vibrations ranging from 5 to 30 kHz can be generated using piezo crystal vibration elements [32].

### 2.4. P3 (Schwenk and Striegel, Germany)

This method involves heating the implant until its surface temperature is sufficiently high to induce osteonecrosis at the bone–implant interface. Furthermore, the implant can also be detached using mechanical vibration. The version shown in Figure 4 depicts an induction source (3) placed on the alveolar bone (2) adjacent to the implant (1). Ferromagnetic material (4) would be embedded within the implant. In this version of the device, the implant is heated by eddy current losses primarily within the ferromagnetic material. As shown in Figure 5, an alternative version of the invention contains an energy source (30) that can be inserted into the internal cavity of the implant [49].

The application of heat in the range of 40–60 °C, preferably around 50 °C, is required. The energy sources are more variable than those described in Patent 1. The mentioned energy sources include high-frequency sound energy (in particular, ultrasonic energy), heat energy or heat radiation energy, magnetic energy (as in induction energy and electromagnetic energy), and, in particular, light or infrared energy, preferably lasers. For better use of induction energy, the incorporation of ferromagnetic materials into implants has been suggested. The ferromagnetic material could be embedded in the “form of layers, rings, or strips”, especially near the implant surface. Additionally, a material with good thermal conductivity could be integrated into the implant in defined geometrical forms depending on the implant geometry.

Ultrasonic sound can be used as mechanical energy to loosen the implant from the bone through vibrations and as a source of thermal energy to heat implants.

The device seems to require at least one sensor (7). The controlling device (15) is connected via a cable (72) to the sensor (7), which may be placed, for example, between the jawbone (2) and the gums or lips (8) in the vicinity of the dental implant (1). The sensor (7) monitors the temperature and/or intensity of the energy supplied by the energy source—(3) in Figure 4 and (30) in Figure 5—to the dental implant. The sensor (7), via the control device (15), can provide feedback regarding the process to the operator, who can stop the process once sufficient thermal destruction of the tissue is achieved [49].

### 2.5. P4 (Petersen and Cattaneo, Denmark)

This device and method (Figure 6) were developed for friction-reducing and bone-saving insertion, as well as the atraumatic loosening and removal of dental implants from the alveolar bone of the patient.

The central idea of this invention is loosening the osseointegrated threaded dental implant through the application of ultrasonic vibrations. The ultrasound vibration actuator generates an ultrasonic vibration with a frequency ranging from 20 to 50 kHz, preferably between 24 and 36 kHz. The vibrations “would be able to disrupt the alveolus bone-to-dental implant contact by the principle of fatigue failure of the bone directly in contact with the dental implant”, without damaging the more distant parts of the alveolar bone or soft tissues, such as the blood vessels, nerves, or gingiva.

The apparatus consists of an ultrasound vibration actuator, which is a handpiece structurally connected to a handpiece head. A rigid and firm connection between the handpiece and the implant is necessary for transmitting ultrasonic vibrations. This can be ensured by connecting devices (“fitting means”) that fit to the coronal, external (AD1), and/or internal (AD2) geometry of the respective implant system and ensuring precise force closure with the holding device (F) in the head of the ultrasound-generating handpiece. The ultrasonic vibration force is limited to rapid clockwise and counterclockwise movements rotating around the longitudinal axis of the dental implant. The area of intervention is cooled by irrigation with a saline solution during the application of ultrasonic vibration. Both measures intend to serve to reduce unwanted damage to the surrounding bone by the generation of heat.

This invention contains a motion sensor, control unit, and user interface. The sensor detects the change in the counterclockwise torque when the dental implant has been loosened, which can be used to control the ultrasonic vibration actuator. The ultrasonic vibration force can be varied by changing the amplitude, frequency, or direction of the vibration. The vibration displacement may be adjusted to the desired level by the user via “a turning knob”.

Predefined vibration patterns may be selectively activated to reduce the vibrations to a minimum within the actual clinical situation.

Another feature of the disclosed method is the use of ultrasonic vibration while inserting/screwing the implant into the prepared cylindrical hole within the bone. The vibrations are intended to reduce the friction between the implant and bone, allowing the drilling of smaller holes in the bone, thereby conserving bone. This feature of the invention is not discussed in this review, as it helps only with implant placement and not removal [41].

### 2.6. P5 (Winnen, Germany)

The fifth patent involves tempering the implant and controlling the heating/cooling process based on information from a database (Figure 7). It is aimed at the formation of a minimally thick layer of denatured bone at the bone–implant interface, just sufficient for osseodisintegration and allowing atraumatic implant removal and maximum conservation of the peri-implant bone [50] (p. 1).

The temperature is produced by a heating/cooling device (10), which can be connected to the inner geometry of the implant (A, B, C) via a coupling device (D, E, F). A temperature sensor can optionally measure the temperature of the connecting device and the inner surface of the implant; however, it can also be omitted in view of the database. A controlling device (20) can be included that features an input device (30) and a database. The database contains information on the amount of energy necessary to induce osseodisintegration of an implant with defined physical properties in defined clinical situations. It can also provide data on the connecting device used to limit the trauma to the surrounding bone to a thin layer.

The input device enables the operator to enter data such as the physical properties of the implant and the specific clinical situation.

The utility of this method can be further enhanced by using it with a dental implant design having a channel (12) along the longitudinal axis of the implant to improve uniform heating and/or cooling. E and F as shown above are exemplary connecting devices for the modified implants [53].

## 3. Comparison of the Methods

The technical approached of each of the five methods described here differ, but they also show significant similarities.

### 3.1. Energy Application

All these devices apply thermal, mechanical, or a combination of these energies to the implant. P1 and P5 rely exclusively on the application of temperature without the use of mechanical forces. P1 involves the application of only heat, whereas P5 involves the application of heat and/or cold. P2 and P3 use a combination of thermal and mechanical approaches, whereas, in P4, the approach is purely mechanical and based on ultrasound. Twelve characteristics of these methods are summarized in Table 1.

### 3.2. Connection

If these methods are used in the oral cavity, each device can include a dental handpiece that is brought into the oral cavity and has a backend connected to other device components. Access to the implant is via its coronal surface, which is the only part that is directly accessible. The head of the handpiece can, therefore, contact the implant to allow the transfer of energy through the “connecting (or coupling) device”. Such a connection ensures a defined and reproducible process. However, P3 also provides an alternative option of applying electromagnetic energy through the adjacent alveolar bone. For the thermal approach, it is ideal to have a sufficient contact surface between the three components (handpiece, connecting device, and implant), and, for the mechanical approach, the connection must be very rigid. For P1 and P5, there can be a point, line, or surface contact between the connecting device and the interior surface of the implant, to modulate the heat conduction.

### 3.3. Process Control

All patents describe that the regulation of the process can occur with the help of at least one sensor. In P5, this function of the sensor can be replaced by information obtained from a database. The sensors measure the energy transferred into the implant (P2, P5), the temperature of the implant (P1, P3, P5), the temperature of the surrounding tissues (P1, P3), or the mobility of the implant (P4). Problems related to measuring the implant temperature are discussed in Section 4.1. All systems have a controlling device with a processor to regulate the application of energy according to the measurements of the sensor. P1 is the only device featuring an alarm. An input device is present in P1 and P5. In P1, the intended temperature and heating time can be entered, while, in P5, the implant data and some individual patient data can be entered into the device.

Furthermore, a database only exists in P5. P4 discloses a “control box” without explaining how this operates and where this box provides predefined vibration patterns. The operator must make the decision with regard to the most appropriate vibration pattern to select for each clinical situation. In contrast, P5 provides a database that can contain implant information and information about the amount of energy that is ideal to remove each specific implant type. For example, the database can include a list of implants by different manufacturers and their characteristics, including implant material, outer form (including length and diameter), inner form, thermal conductivity, and thermal capacity. It can also provide information on the specific thermal conductivity of connecting devices. In the authors’ opinion, a database is essential for the predictable and precise function and control of these systems. This is discussed further in Section 4.1.

The high congruency among the features of these five methods suggests that these features will be useful for any functioning device using energy input for implant removal. In contrast, if certain features are not integrated, it could cause the method to fail; this point is elaborated in Section 4.

## 4. Specific Methodological Challenges

### 4.1. Challenges Related to Temperature-Focused Approaches

Temperature application is the favored approach for the removal of failed implants in four of the five described methods. P2 and P5 involve the use of heat and/or cold application, whereas P2 also focuses on the use of cold application. In P4, although the mechanical forces cause the generation of heat, this effect is unintended.

There is no literature available on the use of these methods in MEDLINE (PubMed) or the Cochrane Library. However, case reports on the application of heat to implants to facilitate their explantation are based on the same concept. Cunliffe and Barclay described heating with electrosurgery for implant removal [44] and continued using the method experimentally [54]. This procedure was based on the clinical method of “thermo-explantation”. Twenty implants in 20 patients were heated using an ultrahigh-frequency electrosurgical device, which significantly reduced the explantation torque after 2 weeks [45]. A CO_2_ laser has also been used to induce thermo-necrosis at the bone–implant interface of osseointegrated implants [46]. All implants could be removed with an explantation torque of approximately 35 Ncm 1 or 2 weeks following the treatment [44,45,46]. 

The primary objective of all modern implant removal methods is minimizing the invasiveness of the procedure. Temperature and its application time are critical to the biophysical induction of intentional osseodisintegration. Widespread bone necrosis must be avoided since it can lead to substantial bone loss and delayed healing [35,36]. Therefore, the control of the temperature at the bone–implant interface is the primary concern in all these methods. It is essential to precisely temper all parts of the implant surface to a defined temperature for a defined duration [35,36,37,38,39,40] to ensure that only a thin layer of bone degenerates and healing occurs quickly and without complications. Heating implants to achieve osseodisintegration without a precise and reliable method of temperature control is problematic due to the risk of widespread necrosis.

Gungormus and Erbasar are currently studying thermal necrosis-aided implant removal using electrosurgery [55]. They performed a three-dimensional finite element analysis of the transient heat transfer in dental implants to identify the optimal energy and time settings for the optimal amount of bone necrosis. They found that even brief contact with electrosurgical probes could drastically increase the implant temperature. They consider the thermal necrosis-aided approach promising and are planning systematic in vivo studies [55]. An in vitro animal study was carried out with the same intention. In both tested settings, the bone was charred around the removed implants, indicating severe damage even in the deeper layers of the bone [56]. Hence, it is evident that there is no appropriate method to regulate the temperature so far.

A sensor measuring the temperature of the implant is a reasonable approach for the regulation of the process. The outer surface of the osseointegrated dental implant is embedded in bone and is inaccessible. The thickness of the surrounding bone varies considerably. Therefore, it is impossible to accurately measure the surface temperature of the implant through the bone. In in vitro experiments, thermocouples inserted into holes drilled into the bone have been used to accurately measure the temperature in different regions of the implants [57]. However, this method is clinically unacceptable. Only the coronal and inner surfaces of the implant, i.e., the contact surface to the mesio-structure, are directly accessible. The temperature of a functional implant can only be measured from these areas. However, the temperature varies significantly across different parts of the implant (crestal, middle, and apical) when energy is applied coronally [57].

Implant designs vary significantly, with implant lengths ranging from 6 to 16 mm and diameters from less than 3 mm to approximately 6 mm [58]. The inner geometry also varies significantly with regard to the differences in the inner or outer connection [59], in addition to differences in the length and position of the screw thread (channel). It has been shown that the time taken to heat a dental implant embedded in the bone can double depending on the implant geometry [55]. Implants are increasingly being designed using different materials with divergent thermal properties. Due to these factors, measuring the implant temperature at its coronal or inner surface is not sufficient to evaluate thermal changes at all points on the outer surface of different implants. However, it is the decisive parameter for process regulation in P1, P2, and P3 [32,41,48].

P1 measures the temperature of the implant through direct contact or infrared radiation at the coronal or inner surface of the implant or the adjacent gingiva [48]. It also recommends the entry of the necessary process time and temperature by the operator. However, the question remains unanswered: what are the optimal temperature and duration for the application of heat? It is clear that these parameters vary considerably with different implant designs [55,57].

P2 also features a temperature sensor. It is positioned in the grip head of the handpiece and measures the temperature of the grip head. The circulation of the cooling or heating media is regulated on the basis of these measurements. Similar to P1, this sensor is also unable to precisely control the temperature on the outer surfaces of different implants.

P3 recommends placing the temperature sensor (7, Figure 2) in the vestibule adjacent to the implant. Correct measurement of the implant surface temperature would not be possible due to the varying dimensions of the bone between the implant and the sensor.

P4 contains a sensor that only measures the mobility of the implant. It does not measure the temperature.

P5 has an optional sensor that measures the temperature on the inner surface of the implant. As with the other methods, it is not capable of measuring the temperature on the outer surface of different implants. It can, however, be used to compare the application of thermal energy to the implant to a specific value from the database to ensure the correct function of the system.

Therefore, temperature measurements proposed in P1, P2, P3, and P5 alone are not suitable for controlling the process of biophysical osseodisintegration using heat or cold. The open questions regarding the process parameters cannot be answered in this manner [48,50,55,56].

According to the rules of thermodynamics, the amount of contact between the heat source and the implant influences the heating process. With thermal conduction, a larger contact surface leads to faster thermal transfer. In contrast, with an electrosurgical probe, a smaller contact surface results in more rapid heating of the implant [55]. This supports the assumption that, according to the energy source, specific connections must be used to ensure a defined and reproducible process.

Since the framework conditions to temper enossal implants are similar (implant in the bone of approximately 36 °C), important variables are the physical data of the implant (material, inner and outer geometry), the type of energy applied, and the connecting device. In P5, data such as these can be entered into the controlling unit. The process parameters for the tempering of different implants can be measured in vitro, modeled using a computer simulation, and stored in a database. Therefore, process control via a database is a solution to precisely match the required energy with the respective implant type.

### 4.2. Challenges Related to the Mechanical Approach

The mechanical approach used in P2, P3, and P4 could successfully disrupt the implant–bone connection with fatigue. In P3, the use of mechanical forces is optional. Furthermore, the introduction of mechanical energy into the implant also increases its temperature.

A preliminary experiment was carried out to test the device and method described in P4; Brånemark Mark III dental implants were embedded in the following materials [41]:Snow white plaster No. 2, Kerr.A bone-like plastic material (polyurethane foam).Fresh bovine tibia bone.Hard epoxy-bone cementum (Refobacin^®^, polymethyl acrylate, and polymethyl methacrylate).

Cylindrical hollow superstructures of three different lengths were mounted on the dental implant platform using stainless steel screws. Ultrasonic vibration combined with a light counterclockwise rotation torque was applied to the implants. The implants could be loosened from the plaster, polyurethane foam, and tibia bone. However, the (tibia) bone walls of the implant bed were discolored after the removal of the implant, indicating a substantial increase in the temperature despite the use of water irrigation during the loosening procedure, and the implants were not osseointegrated in the test materials, just “friction locked”. With fully osseointegrated implants, the time taken to perform the procedure would be much longer and, consequently, more heat would be generated, causing likely unnecessary trauma in the patient. According to Petersen and Cattaneo, “to which extent such heating will affect the vitality of the surrounding bone and neighboring structures can only be tested in animal experiments” [41].

P2 also refers to the generation of heat in an implant through the transmission of vibration energy. Negative effects on the bone are prevented by simultaneously cooling the implant with CO_2_ or another cooling agent. The combination of heat generation in a precooled or simultaneously cooled implant makes the tempering of the implant surface even more complex. It cannot be adequately controlled by a temperature sensor placed in the handpiece grip head, as intended in P2. P5 also has the option of applying ultrasonic energy to the implant. However, with P5, the intention is to purely heat the implant. A possible mechanical component would be a side-effect.

Ideally, one would know the temperature at all points of the implant surface since all vibrations applied to the implants generate heat, as with the temperature-focused methods. As described earlier, it is currently difficult to perform such measurements accurately for the regulation of the osseodisintegration process clinically. Furthermore, it should also be considered that the application of vibrations to the osseointegrated implant may cause discomfort to the patient.

### 4.3. Challenges Related to Implant Materials (P3)

P3 suggests embedding ferromagnetic materials or materials with high thermal conductivity into implants. Implants with such a composition have not yet been developed. Therefore, this technique could benefit only future generations of implants. Production of implants with a multi-material composition would be technically complicated and more expensive. The materials in multilayered implants would further expand and shrink at different temperatures (for example, when eating hot food), causing mechanical stress, perhaps even causing structural weakening of the implant. Furthermore, electrochemical processes occurring at the interface of different metals can cause corrosion, which could compromise osseointegration and reduce the implant longevity [60,61,62,63,64]. Magnetic resonance examination of the patients could also be hindered [65]. The authors of P3 recognized some of these issues themselves and recommended “minimal embedding of ferromagnetic materials”. Further research on the possible biological repercussions of adding ferromagnetic materials in dental implants is necessary.

### 4.4. Challenges Related to Implant Design (P5)

P5 (optionally) describes an implant that “comprises a channel extending along the longitudinal axis of the implant over a distance of … up to 98% of the length of the implant” [66]. In contrast to the multi-material approach in P3, implants as described in P5 can be made rather easily, e.g., by modifying existing implants by drilling the channel. Thus, unlike P3, the production of implants with channels would be significantly less technically complicated and may not be very expensive. Screw channels and cavities for the reception of abutments with substantial diameters have been an essential feature of implant design for decades. Reduced wall thickness is mostly compatible with implant function. The extended channel has a smaller diameter than the abutment, leaving more substantial walls. It is located in the apical part of the implant, distant from the mechanical stress induced by the abutment in function. Therefore, it should not affect the mechanical properties of the implant and does not have an electrochemical influence.

Preliminary calculations and ongoing research [40,67] confirm that the method described in P5 is clinically applicable with conventional implants. It is particularly suitable for implants containing a long channel along their rotational axis, facilitating more homogeneous heating (or cooling) of the implants and, consequently, minimizing the extent of the denatured bone.

## 5. Summary

The five techniques of minimally invasive implant removal presented here are based on the concept of loosening the implant–bone connection through the input of energy, which seems promising. However, regulation of these processes is necessary to achieve implant–bone separation and avoid deeper damage. Temperature measurement with the described sensors alone may not accurately indicate the temperature at the bone–implant interface, especially at all implant surfaces; therefore, it seems insufficient for the precise regulation of the process. Since the framework conditions for inducing thermal changes in an enossal implant are similar, the most important variables include the physical characteristics of the implant. Process parameters for the induction of thermal changes in different implants can be measured in vitro, modeled using a computer simulation, checked in animal models, and then stored in a database. Therefore, process control via a database will be more precise than when only relying on a sensor-based approach and could also be adapted for cementless medical implants.

Future research will further optimize the methods discussed in this review. For example, optimal temperatures and time values required to induce osseodisintegration in vivo with reproducible processes can be optimized for a variety of the most commonly used implants. The healing time required before the placement of a new implant and the possible risks associated with the temperature application method warrant further research. Together with the methods described in P1 through P5, this will pave the road toward making the next generation of implants and medical devices that facilitate the safe and effective removal of enossal implants, while minimizing trauma to the patient.

## 6. Conclusions

Loosening the implant–bone connection through the application of energy seems promising.Precise control over the processes is necessary to avoid deeper damage.Temperature measurement using sensors is insufficient for process control.The physical characteristics of the implant are decisive variables.Process parameters for different implants can be measured and stored in a database.Process control via a database can tailor energy output to a selected implant and minimize trauma to the patient.This could allow reversibility of osseointegration.Similar methods could be adapted for cementless medical implants.Further research in this field will further advance the field of osseodisintegrative methods.

## 7. Patents

The “device for the controlled removal of osseointegrated implants and improved dental implants” described as Patent 5 in the manuscript was patented by the author Rolf Winnen.

## Figures and Tables

**Figure 1 materials-14-07829-f001:**
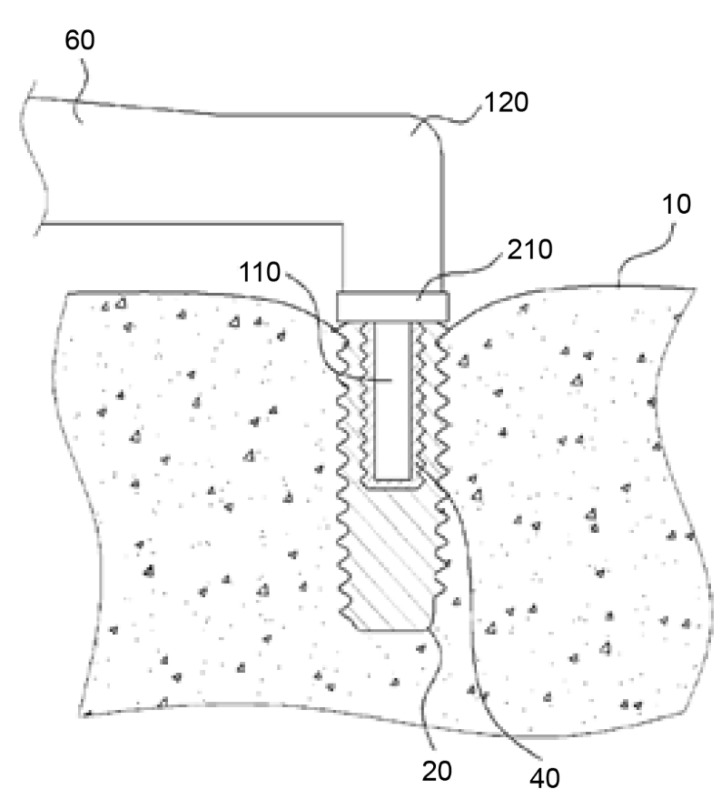
Handpiece (60) with a heating unit (120), temperature sensor (210), and connecting device (110). The heating unit provides heat between 40 °C and 1000 °C. The temperature sensor measures the temperature of the implant (20) and surrounding tissues (10). Source: [51].

**Figure 2 materials-14-07829-f002:**
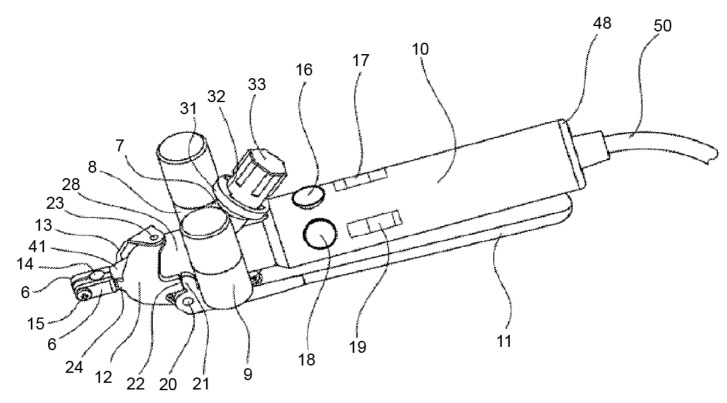
This image shows the heating or cooling module (7), vibration module (8), and hammering module (9). Source: [32].

**Figure 3 materials-14-07829-f003:**
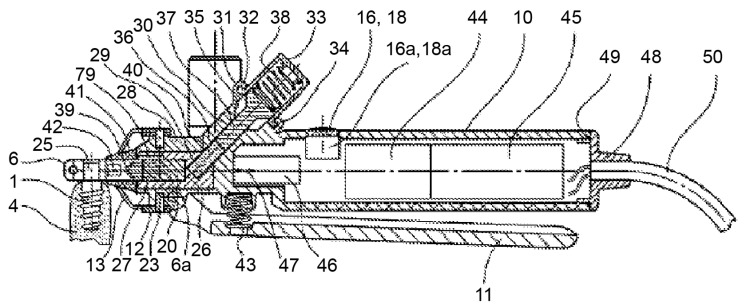
The device consists of a handpiece containing a grip head (6), handle (10), and a secondary clamping system composed of a clamping lever (11) and clamping jaws (12). They can exert clamping forces to the implant (1) in the bone (4). Source: [32].

**Figure 4 materials-14-07829-f004:**
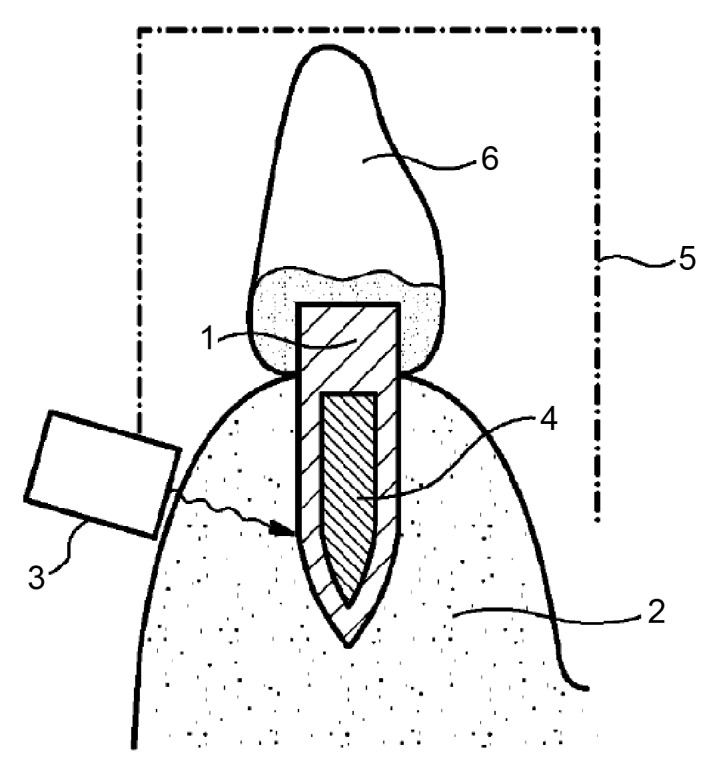
An induction source (3) can be placed on the alveolar bone (2) adjacent to the implant (1). Ferromagnetic material (4) is embedded within the implant. The implant is heated by eddy current losses, working primarily within the ferromagnetic material. Source: [49].

**Figure 5 materials-14-07829-f005:**
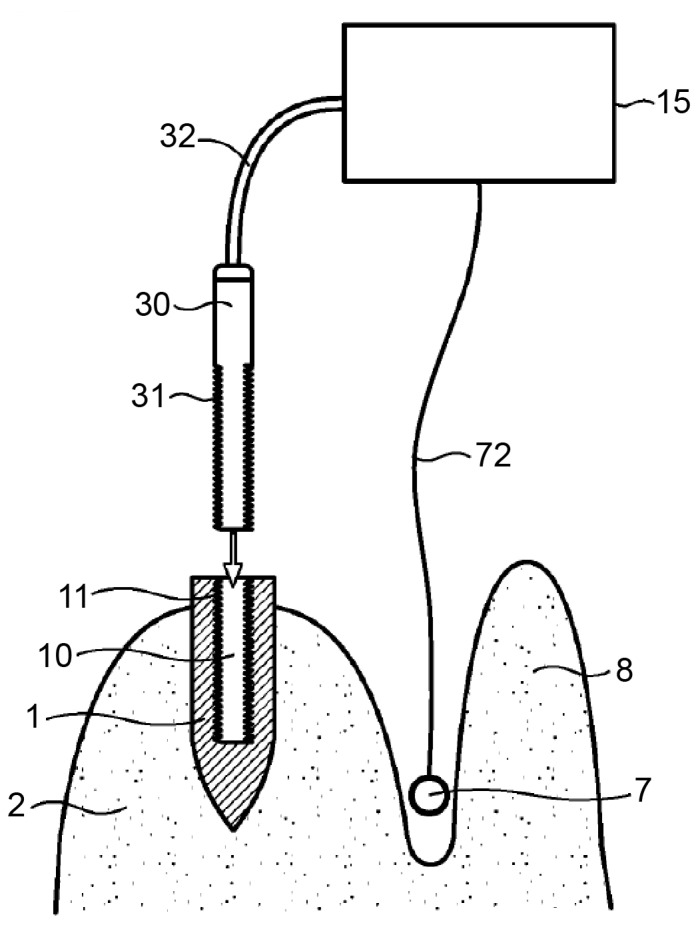
An alternative version of the invention contains an energy source (30) that can be inserted into the internal cavity of the implant. The control device (15) is connected via a sensor line (72) to a sensor (7), which can be placed between the jawbone (2) and the lip (8) in the vicinity of the dental implant (1) to be treated. Source: [49].

**Figure 6 materials-14-07829-f006:**
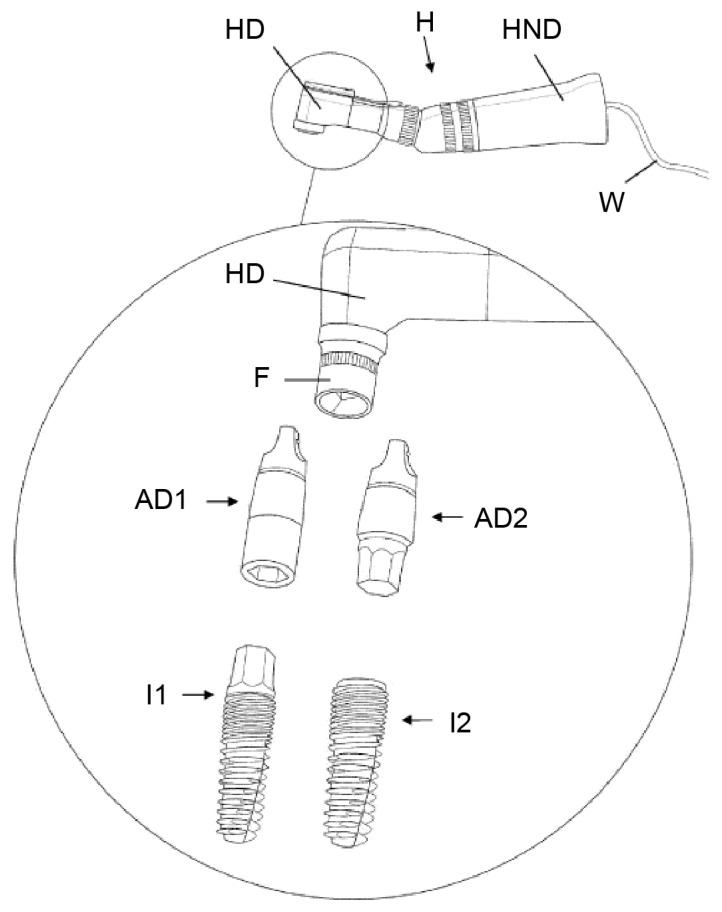
An ultrasound vibration actuator in a handpiece (HND) with a handpiece head (HD) and holding device (F) are shown. AD1/2 ensures a rigid and firm connection between the handpiece and implants I1/2 to transfer the ultrasonic energy into the implants. Source: [41].

**Figure 7 materials-14-07829-f007:**
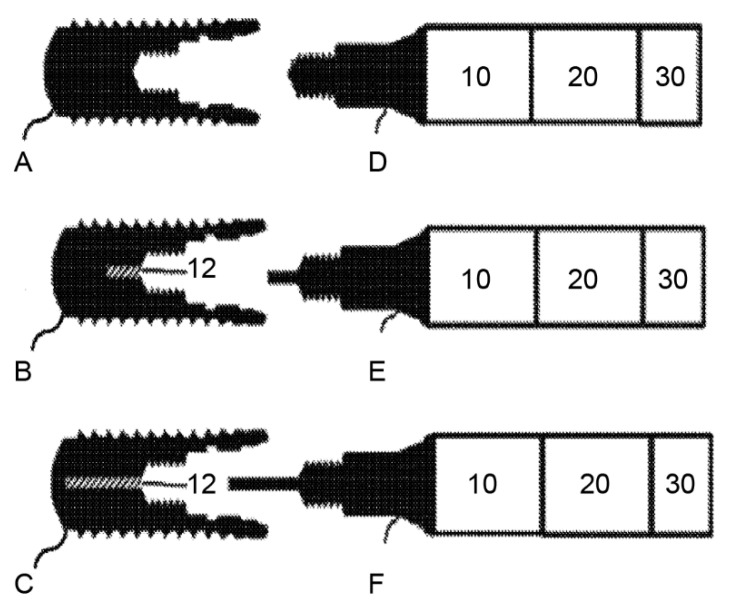
Implants (**A**–**C**) on the left side and the corresponding coupling devices (**D**–**F**) on the right side (10 refers to the heating device, 20 refers to the controlling device, and 30 refers to the input device). They can be connected to the inner geometry of the implants via the coupling devices. Implants B and C contain a channel to heat and cool the implants more homogenously. Source: [53].

**Table 1 materials-14-07829-t001:** Comparison of the devices.

No.	Device Characteristics	P1 (Lee)	P2 (Braegger et al.)	P3 (Schwenk and Striegel)	P4 (Peterson and Cattaneo)	P5 (Winnen)
	**Energy application**					
1	Heating	+	+ optional	+	+ (unintended)	+
2	Cooling	−	+	−	−	+ (optional)
3	Vibration	−	+	+ optional	+	−
	**Connection**					
4	Handpiece	+	+	+	+	+
5	Connecting device	+	+	+	+	+
6	Coronal access	+	+	+ optional	+	+
	**Process control**					
7	Sensor(s)	+	+	+	+ (mechanical)	+(optional)
8	Controller	+	+	+	+	+
9	Processor	+	+ (mechanical)	+	+	+
10	Database	−	−	−	+ (“control box“)	+
11	Input	+	−	−	−	+
12	Alarm	+	−	−	−	−

+: a feature of the device; −: not a feature of the device.

## Data Availability

Not applicable.

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
