# Peer review of "Reversal of Osseointegration as a Novel Perspective for the Removal of Failed Dental Implants: A Review of Five Patented Methods"

_materials, 2021, doi:10.3390/ma14247829_

Round 1

Reviewer 1 Report

The authors conducted a nice review on an interesting and novel topic and I believe that, after improving some minor points, the article could be of interest for the readers.

1) Line 53

The authors should add these papers relevant to the concept of the interaction between systemic conditions and peri-implantitis (Prevalence of peri-implant diseases among an Italian population of patients with metabolic syndrome: A cross-sectional study.

Papi P, Di Murro B, Pranno N, Bisogni V, Saracino V, Letizia C, Polimeni A, Pompa G.J Periodontol. 2019 Dec;90(12):1374-1382. doi: 10.1002/JPER.19-0077. Epub 2019 Aug 3.PMID: 31328267   Obesity/Metabolic Syndrome and Diabetes Mellitus on Peri-implantitis. de Oliveira PGFP, Bonfante EA, Bergamo ETP, de Souza SLS, Riella L, Torroni A, Benalcazar Jalkh EB, Witek L, Lopez CD, Zambuzzi WF, Coelho PG.Trends Endocrinol Metab. 2020 Aug;31(8):596-610. doi: 10.1016/j.tem.2020.05.005. Epub 2020 Jun 23.   Association between diabetes mellitus/hyperglycaemia and peri-implant diseases: Systematic review and meta-analysis. Monje A, Catena A, Borgnakke WS.J Clin Periodontol. 2017 Jun;44(6):636-648. doi: 10.1111/jcpe.12724.   2) Line 105-129 can be shortened, summarized and placed in the introduction section   3) please discuss how many results can be found in the patent database for implant removal, apart from those included in this review and not yet scientifically validated   

Author Response

1) Line 53

The authors should add these papers relevant to the concept of the interaction between systemic conditions and peri-implantitis (Prevalence of peri-implant diseases among an Italian population of patients with metabolic syndrome: A cross-sectional study.

Papi P, Di Murro B, Pranno N, Bisogni V, Saracino V, Letizia C, Polimeni A, Pompa G.J Periodontol. 2019 Dec;90(12):1374-1382. doi: 10.1002/JPER.19-0077 . Epub 2019 Aug 3.PMID: 31328267   Obesity/Metabolic Syndrome and Diabetes Mellitus on Peri-implantitis. de Oliveira PGFP, Bonfante EA, Bergamo ETP, de Souza SLS, Riella L, Torroni A, Benalcazar Jalkh EB, Witek L, Lopez CD, Zambuzzi WF, Coelho PG.Trends Endocrinol Metab. 2020 Aug;31(8):596-610. doi: 10.1016/j.tem.2020.05.005 . Epub 2020 Jun 23.   Association between diabetes mellitus/hyperglycaemia and peri-implant diseases: Systematic review and meta-analysis. Monje A, Catena A, Borgnakke WS.J Clin Periodontol. 2017 Jun;44(6):636-648. doi: 10.1111/jcpe.12724 .  

Response: Thank you for your valuable suggestion. The suggested papers have been added to the manuscript as per your suggestion (Lines 60-61).

2) Line 105-129 (old version) can be shortened, summarized and placed in the introduction section  

Response: Thank you for your valuable suggestion. As per the suggestion, we have tried to shorten and summarize the paragraphs (lines 105–129). The paragraphs have been placed in the Introduction section.

3) please discuss how many results can be found in the patent database for implant removal, apart from those included in this review and not yet scientifically validated

Revised text (Page 3, Lines 135–144):

The patents were selected based on the results of the search conducted by the European Patent Organization (EPO) in the Patentscope database during the patent application process of Patent 5 (see below). The searches were conducted according to “search procedure and strategy” described in Part B of the Guidelines for Examination in the EPO [40]. The purpose of the search strategy was to find all similar patents, in this case, devices for the removal of enossal implants through the application of energy. Since EPO is renowned for its diligence in this respect, this goal was achieved.

Reviewer 2 Report

The topic of the present study, comparing five methods for the removal of failed dental implants, is very interesting, especially considering the clinical implications.

The narrative review is well structured and written. 

The introduction is appropriate to the topic and relevant references have been added.

The authors described in detail the methods for the removal of failed implants and related devices and comparisons among them were sufficiently discussed. 

No speculations on reccommendations were given about specific clinical indications for each methods.

The only suggestion the reviewer could give may be to specify in the text the methods concerning search strategy (sources, criteria, timing, etc.) and study inclusion criteria.

Author Response

The only suggestion the reviewer could give may be to specify in the text the methods concerning search strategy (sources, criteria, timing, etc.) and study inclusion criteria.

Revised text (Page 3, Lines 137–144):

The patents were selected based on the results of the search conducted by the European Patent Organization (EPO) in the Patentscope database during the patent application process of Patent 5 (see below). The searches were conducted according to “search procedure and strategy” described in Part B of the Guidelines for Examination in the EPO [40]. The purpose of the search strategy was to find all similar patents, in this case, devices for the removal of enossal implants through the application of energy. Since EPO is renowned for its diligence in this respect, this goal was achieved.

Reviewer 3 Report

I'm very glad to become acquainted with such interesting research. Hope that my review will help to improve the quality of the article. The authors made significant work.
The manuscript is dedicated to the analysis of methods to remove failed dental implants. In the article five patents were studied, advantages and disadvantages of all of them are studied. The results can be useful for dental clinics.
The work is well organized and I have no remarks.

Author Response

…, there are some related biological issues that I considered have not been enough revised and discussed. I recommend authors always keep in mind that implants are meant for supporting structures in live tissues, so some topics should also be revised and discussed. Some of these topics could be:

  1. Authors should give indications for wanting to disintegrate a dental implant. Very general concepts have been revised in the introduction. They should be more specific.

Response: Thank you for your comment. We have added the indications for disintegrating the implant. The criteria and indications for disintegrating the implant are not completely clear yet. If a microinvasive implant removal technique becomes available, it may lead to a change in the indications for implant removal. We have made the following changes in the revised manuscript:

Revised text:

(Page 1, Lines 40-42)

Peri-implantitis (81.9%), implant mal-positioning (2%–14%, correlated with the expe-rience of the surgeon), and implant fracture (1%) are frequent causes of implant failure [1] and are indications for implant removal.

(Page 2, Lines 50–51)

The criteria for the preservation or removal of implants are not yet clearly defined and remain a subject requiring further research.

(Page 3, Lines 102–107 and 130-132)

The implants can undergo osseointegration even in the presence of malpositioning or implant fracture as indications for implant removal being the indication. The osseoin-tegrated part of the implant can comprise more than the apical 4 mm when moderate or medium peri-implantitis is the indication. In such cases, the implant must initially be separated from the bone to an extent such that CTRT becomes possible. The classical approach involves resection using a trephine bur.

  1. Discuss the biological repercussion of adding ferromagnetic materials in dental implants. Would it affect the long-term osseointegration of implants? And the immediate osseointegration? The authors should revise this issue, and if no answer can be given, it may be a good starting point for future research.

Response: Thank you for your comment. According to the literature, electrochemical processes occurring at the interface of different metals can cause corrosion. This could compromise the osseointegration and reduce the longevity of the implant.

Revised text (Page 15, Lines 590–592):

Further, electrochemical processes occurring at the interface of different metals can cause corrosion, which could compromise osseointegration and reduce the implant longevity [60–64].

  1. Discuss the mechanical repercussion of changing the design of implants so thermal changes can be applied or measured feasibly. Would it affect the long-term biomechanical prognosis of implants, or would they become weaker and might break?

Response:

Revised text (Page 15, Lines 606–613):

Screw channels and cavities for the reception of abutments with substantial diameters have been an essential feature of implant design for decades. Reduced wall thickness is mostly compatible with implant function. The extended channel has a smaller diameter, leaving more substantial walls. It is located in the apical part of the implant, distant from the mechanical stress induced by the abutment in function. Therefore, it may not affect the mechanical properties of the implant and does not have an electrochemical influence.

  1. Thermal changes are described as a safe method to loosen the osseointegrated implants, but the manuscript should discuss the biological repercussions of the thermal necrosis caused when performing such procedures. 4.1 Can thermal necrosis be controlled? 4.2 How is it going to heal? 4.3 How long will it take this healing so a new implant can be placed? 4.4 Any risk of causing further disease when applying so high temperatures to the bone?

Response: Thank you for your questions. We have added the following points to the manuscript to address your questions.

Revised text:

(Page 13, Lines 467–476)

The primary objective of all modern implant removal methods is minimizing the invasiveness of the procedure. Temperature and its application time are critical to the bio-physical induction of intentional osseodisintegration. Widespread bone necrosis must be avoided since it can lead to substantial bone loss and delayed healing [45,46]. There-fore, the control of the temperature at the bone-implant interface is the primary concern in all these methods. It is essential to precisely temper all parts of the implant sur-face to a defined temperature for a defined duration [45–50] to ensure that only a thin layer of bone degenerates and healing occurs quickly and without complications. Heating implants to achieve osseodisintegration without a precise and reliable method of temperature control is problematic due to the risk of widespread necrosis.

(Page 16, Lines 637–639)

The healing time required before the placement of a new implant and the possible risks associated with the temperature application method warrant further research.

  1. It isn't easy to differentiate when reading the discussion or the conclusions. Besides, in my opinion, conclusions should be more precise and more concise. Using a dot list would be an excellent way to get it. 

Response: Thank you for your valuable suggestion. As per your suggestion, we have created a bulleted list to make the conclusion section more concise and precise.

Revised text (Page 16, Lines 648–659):

  • Loosening the implant-bone connection through the application of energy seems promising.
  • Precise control over the processes is necessary to avoid deeper damage.
  • Temperature measurement using sensors is insufficient for process control.
  • The physical characteristics of the implant are decisive variables.
  • Process parameters for different implants can be measured and stored in a database.
  • Process control via a database could act as a solution.
  • It could allow reversibility of osseointegration.
  • The method could be adapted for cementless medical implants.
  • Further research in this field is necessary in the future.

Reviewer 4 Report

Congratulations to the authors for having written such a manuscript. It may be the beginning of many research lines in the future. Various novel methods to lose implant integration have been described from a technical point of view. However, there are some related biological issues that I considered have not been enough revised and discussed. I recommend authors always keep in mind that implants are meant for supporting structures in live tissues, so some topics should also be revised and discussed. Some of these topics could be:

  1. Authors should give indications for wanting to disintegrate a dental implant. Very general concepts have been revised in the introduction. They should be more specific.
  2. Discuss the biological repercussion of adding ferromagnetic materials in dental implants. Would it affect the long-term osseointegration of implants? And the immediate osseointegration? The authors should revise this issue, and if no answer can be given, it may be a good starting point for future research.
  3. Discuss the mechanical repercussion of changing the design of implants so thermal changes can be applied or measured feasibly. Would it affect the long-term biomechanical prognosis of implants, or would they become weaker and might break?
  4. Thermal changes are described as a safe method to loosen the osseointegrated implants, but the manuscript should discuss the biological repercussions of the thermal necrosis caused when performing such procedures. Can thermal necrosis be controlled? How is it going to heal? How long will it take this healing so a new implant can be placed? Any risk of causing further disease when applying so high temperatures to the bone?

It isn't easy to differentiate when reading the discussion or the conclusions. Besides, in my opinion, conclusions should be more precise and more concise. Using a dot list would be an excellent way to get it.

Thank the authors for the manuscript. As I said before, it gives osseointegration research new ideas to go on. The document is very well written from a technical point of view, but it would also be interesting to add a biological point of view.
